# Simulation of TiN/Ti Multilayer Coating under the Impact of Multiple Particles Based on Cohesive Element

Zhanwei Yuan [1],*, Wenlong Shi [1], Guangyu He [2], Yan Chai [2], Zhufang Yang [2] and Min Guo [3],*

[1] School of Materials Science and Engineering, Chang'an University, Xi'an 710061, China; 13624824409@163.com
[2] Science and Technology on Plasma Dynamics Laboratory, Air Force Engineering University, Xi'an 710038, China; hegy_22@126.com (G.H.); chaiyan1026@163.com (Y.C.); yangzf1113@126.com (Z.Y.)
[3] Inner Mongolia Metal Material Research Institute, Baotou 014000, China
* Correspondence: yuanzhw@chd.edu.cn or yuanyekingfly@163.com (Z.Y.); guomin-2014@sohu.com (M.G.)

**Abstract:** In order to investigate the damage behavior of TiN/Ti multilayer coating under multi-particle impact and the influence of impact angle on erosion resistance, the ABAQUS 2019 software and the cohesive element technology were used for simulation. The results showed that, during the impact process, the upper surface of the TiN layer that was directly below the impact center was mainly subjected to compressive stress, while the lower surface was subjected to tensile stress. At the impact contact edge, tensile stress appeared on the upper surface of the TiN layer, while compressive stress appeared on the lower surface. The increase in the number of impacts leads to an increase in the maximum S11 stress inside the coating and the maximum displacement of the impact center during the impact process. The plastic damage was greater at the locations with higher strain in the Ti sublayer. During the impact process, severe damage occurred in both the top TiN layer and interface areas, and material failure occurred in the impact area. The increase in impact angle leads to an increase in the plastic strain energy of the entire model after the impact and the maximum S11 stress inside the coating during the impact process.

**Keywords:** TiN/Ti multilayer coatings; multi-particle impact; cohesive element; damage evolution; ABAQUS





## 1. Introduction

When aircrafts (such as military helicopters or transport planes) operate in harsh environments such as deserts and battlefields, especially during taking-off, landing, and low-altitude hovering, a local high-concentration environment enriched in a large amount of sand and dust particles is formed under the action of the rotor downwash airflow [1–3]. A large amount of sand and dust particles are sucked into the engine with the air and then repeatedly impact and rub the engine's working blades at a high speed. The repeated high-speed impact and friction of engine's working blades cause wear and fatigue crack initiation and propagation in the surface material of the blades [4,5], resulting in erosion damage to the blades, which has a negative impact on the reliability and safety of the engine and seriously reduces the service life of the blades [6]. Due to high specific strength, high specific stiffness, low density, and other advantages, titanium alloy is commonly used in aviation engine blades and disks [7]. However, its poor impact wear resistance and low hardness make titanium alloy blades face significant erosion damage problems in environments enriched in sand and dust [8].

A large number of studies have shown that the erosion mechanism of sand and gravel on blades during the erosion process is very complex, which is the coupling of sand and gravel impact and wear processes on materials [9]. The erosion speed, impact angle, and quality of sand and gravel are important factors affecting the degree of erosion damage [10,11]. To solve this problem, the preparation of erosion-resistant coatings on

the blade surface can effectively improve its erosion resistance performance, so as to effectively improve the service life of the engine blade [12–14]. It is generally believed that the preparation of hard coatings is an important means to improve erosion resistance. In recent years, a large number of scholars have studied ZrN, TiN, and other ceramic coatings [15,16]. The high hardness and wear resistance of hard coatings are beneficial for improving erosion resistance, but they can reduce the toughness of the coating, making the impact damage of sand particles on the coating more severe [1]. The toughness of the coating has gained increasing attention [17]. For multilayer coatings with alternating hard ceramic and soft metal layers, their erosion resistance is better than that of single-layer hard coatings [18]. In recent years, in the study of the impact damage characteristics of titanium nitride/titanium (TiN/Ti) multilayer coatings, the main research method is sand and dust erosion testing. Although it is close to the real service environment, it also has the disadvantages of high test cost, long cycle, limited experimental equipment, and weak anti-interference ability of the test to the environment [19,20]. Through simulation methods, it is possible to continuously and dynamically study the overall and local impact processes of multiple particles [21,22]. The common numerical simulation methods include finite element method [23,24], finite difference method [25,26], and particle discrete element method [27,28]. Zhang et al. [29] conducted finite element analysis on the impact process of TiN/Ti multilayer coatings and found that the plastic strain of the matrix decreased with the increase in the number of layers when the thickness of the coating was unchanged. Xu [19] conducted a finite element analysis of the stress distribution in the impact area and found that repeated high stress gradients in the hard layer and at the interface of the binding layer and transition layer were the main reasons for spalling. Zhang et al. [30] simulated the impact characteristics of the multilayer coating and found that the multilayer structure could hinder the propagation of stress waves inside the coating and reduced the stress inside the coating and in the matrix.

In order to study the stress and damage evolution law of multilayer coating under the impact of multiple sand particles, ceramic/metal (TiN/Ti) multilayer coating was selected as the research object. A three-dimensional impact model with cohesive elements was established using ABAQUS finite element software (Version 2019) to improve the accuracy of calculations and reduce computational time. Three particle impacts were used to represent the multi-particle impact process and simultaneously simulate and analyze the influence of the impact angle on the impact process. The multi-particle impact process is represented by three particle impacts, and the influence of the impact angle on the impact process is simulated and analyzed.

## 2. Finite Element Model

### 2.1. Establishment of Geometric Models

The multi-particle impact model of TiN/Ti multilayer coating was established with the ABAQUS software. The coating thickness was 24 μm, the modulation ratio was four, and the total number of coating layers was eight. The coating modulation ratio represented the ratio of TiN layer thickness to Ti layer thickness. In order to simulate the multi-particle impact process, three rigid, spherical particles were set up. According to the particle size analysis of the Taklimakan Desert sand [31], the radius of the rigid sphere was 75 μm. Figure 1 shows the schematic diagram of the geometric model, which adopts a 1/2 symmetric model and has a geometric size of 1000, 500, and 500 μm in length, width, and height, respectively. To improve simulation accuracy and reduce computational complexity, the mesh in the contact area between particles and coatings was densified, while the mesh in other areas was sparser. The plane along the length direction of the coating and matrix in the geometric model was set as a symmetric constraint (YSYMM), and the bottom and other sides of the model were set as full freedom constraints. The impact velocity of the particles was 100 m/s, and the impact angles were set as 15°, 30°, 45°, 60°, 75°, and 90°, respectively, and no contact was set among the three particles. The number of elements in the coating-matrix model is 512,044 (including 296,164 C3D8 elements and 215,880 COH3D8 elements), and

the number of elements in a single particle is 1176 (including 840 C3D8 elements and 336 C3D4 elements). C3D8, COH3D8, and C3D4 are all grid element types in the ABAQUS software. C3D8 represents an eight-node linear hexahedral element, C3D4 represents a four-node linear tetrahedral element, and COH3D8 represents an eight-node three-dimensional cohesive element.

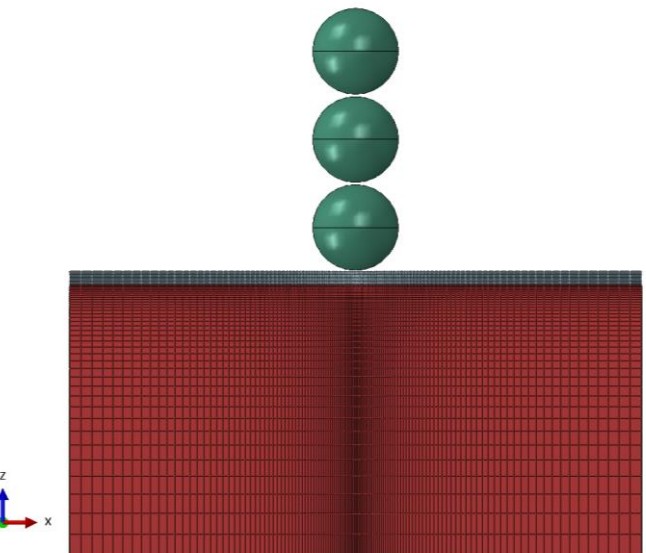

**Figure 1.** Finite element model.

## 2.2. Material Property Settings

In the impact model, TC4 (Ti-6Al-4V) titanium alloy material was selected as the matrix material, and the coating is made of metallic material Ti and ceramic material TiN. Aluminum oxide ($Al_2O_3$) particles were used as sand particles. The basic parameters of the four materials are shown in Table 1.

**Table 1.** Properties of model's material [29].

| Material Name | Ti-6Al-4V | TiN | Ti | $Al_2O_3$ |
|---|---|---|---|---|
| Density (kg·m$^{-3}$) | 4428 | 5400 | 4500 | 3970 |
| Elastic modulus (GPa) | 113.8 | 480 | 110 | 344 |
| Poisson's ratio | 0.34 | 0.27 | 0.33 | 0.2 |

Since TiN is a brittle coating with high hardness, it is assumed to be a complete elastomer in this model. The metal sublayer Ti and the substrate TC4 belong to elastic–plastic materials. The Johnson–Cook (J-C) constitutive model was used to describe the plastic constitutive relationship between the substrate TC4 and the metal sublayer Ti material. This constitutive model is commonly used in high-speed impact and explosion problems and can describe the strain rate-strengthening effect of metals [32]. The expression of the J-C equation is as follows:

$$\sigma_f = (A + B\varepsilon^n)\left[1 + C \ln\left(\frac{\dot{\varepsilon}}{\dot{\varepsilon}^0}\right)\right]\left[1 - \left(\frac{T - T_r}{T_m - T_r}\right)^m\right] \tag{1}$$

where A, B, and C, respectively, represent yield strength, strain hardening coefficient, and strain rate constant; $\varepsilon$ is the equivalent plastic strain; $\dot{\varepsilon}$ is the strain rate; and $\dot{\varepsilon}^0$ is the reference strain rate, usually $1\,s^{-1}$. T, $T_r$, $T_m$, and m represent the current temperature, room temperature, material melting temperature, and temperature sensitivity index, respectively.

Since the temperature in the erosion process is close to room temperature and the effect of temperature on the erosion response is very small, the variable of temperature is

excluded from the simulation process. The J-C constitutive model parameters of the base TC4 titanium alloy and Ti sublayer are shown in Table 2:

**Table 2.** J-C constitutive model parameters of matrix (Ti-6Al-4V) and Ti [29].

| Material Property | Symbol | Ti-6Al-4V | Ti |
|---|---|---|---|
| J-C yield strength | A (MPa) | 1098 | 277 |
| J-C hardening coefficient | B (MPa) | 1092 | 894 |
| J-C strain hardening exponent | n | 0.93 | 0.57 |
| J-C strain rate constant | C | 0.014 | 0.06 |
| J-C reference strain rate | $\dot{\varepsilon}^0$ | 1 | 1 |

*2.3. Cohesive Zone Model*

For hard coating, a bilinear cohesive element model based on the energy failure criterion was used. Zero-thickness cohesive elements were inserted into the TiN layer and the interface between the TiN layer and the Ti layer. The traction separation criterion was used to describe the deformation state between the internal elements of the particles. Figure 2 shows the constitutive relationship of the bilinear cohesive element model. When the cohesive region stress $\sigma$ at the crack tip is equal to the interfacial strength $\sigma_0$, damage occurs, and crack initiation occurs. Under the continuous action of an external load, cracks continue to expand, the stiffness of the material declines, and the bearing capacity weakens.

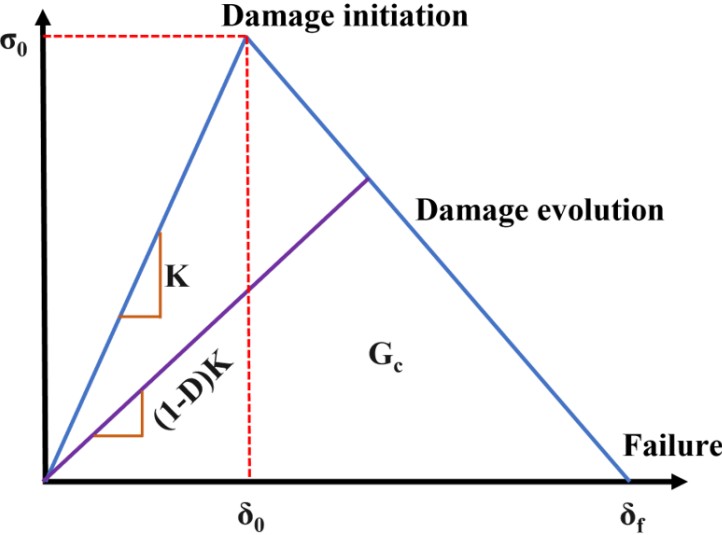

**Figure 2.** Bilinear cohesive model constitutive relations.

## 3. Results and Discussion

Figure 3 shows the morphology of the coating in the impact area after the impact of different numbers of particles. It can be seen that, when the number of impact particles increased from one to four, the top TiN layer changed from having obvious cracks and a small amount of debris to having a large amount of material stripping, and even obvious fractures and debris generation occurred in the second TiN layer. This indicates that the increase in the number of shocks will aggravate the damage to the coating. In the numerical simulation, the increase in the number of shocks will increase the calculation amount and calculation time, and too much debris may cause element distortion, which can affect the accuracy of the simulation. Therefore, in this article, three particle impacts were used to represent the multi-particle impact process in order to study the damage law and energy changes during the multi-particle impact process.

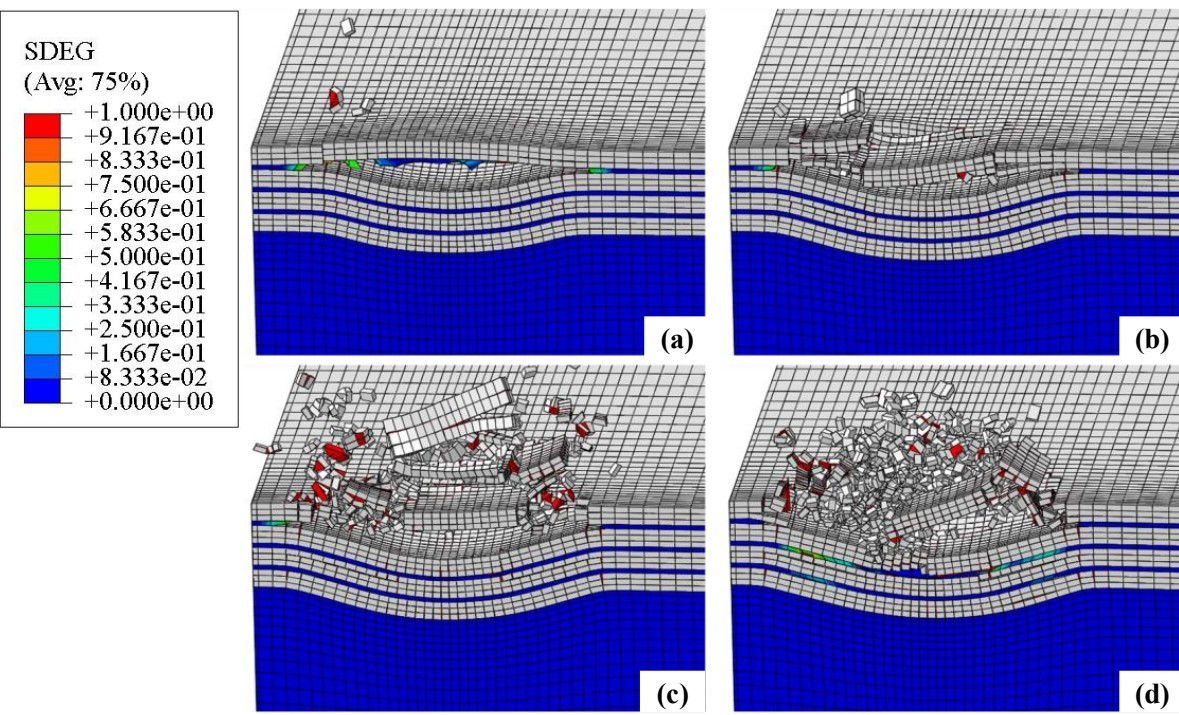

**Figure 3.** Effect of number of impact particles: (**a**) single particle; (**b**) double particle; (**c**) triple particle; and (**d**) quadruple particle.

Figure 4 shows the displacement changes in sand grains and coating impact centers during the impact process. As shown in the figure, each impact is the process of particles pressing down on the coating surface and bouncing back to separate each other. In the figure, the maximum displacements of particles and coating surfaces in each impact are 9.0383, 11.6488, and 12.719 μm, respectively, and the displacements increase continuously during separation. It shows that the deformation and damage to the coating are gradually accumulated and aggravated with the increase in impact time. At the same time, the displacement of the impact center of the coating increases steadily after 1375 ns because serious damage occurs at the impact center, and the material is stripped away from the coating surface.

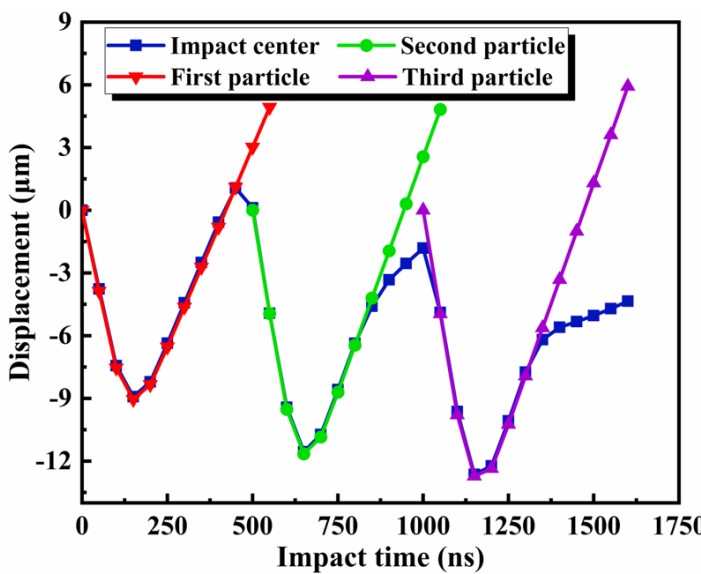

**Figure 4.** Displacements of impact particle and coating impact center at different times.

Figure 5 shows the energy changes in the entire model during the impact process, and it can be seen that there are three significant changes in kinetic energy, elastic potential energy, and plastic strain energy. The initial kinetic energy, including the three particles, is 0.103 mJ. In each impact, as the particles move to the lowest position and bounce off the coating surface, the kinetic energy first decreases and then increases, and the elastic potential energy first increases and then decreases, while the plastic strain energy and the dissipation energy of crack damage first increase and then remain stable. At the lowest position of particle movement, the elastic potential energy, plastic strain energy, and crack damage dissipation energy reach their maximum values. In the kinetic energy consumed after the first impact, plastic strain energy, crack damage dissipation energy, and elastic potential energy accounts for 58.9%, 13.2%, and 20.5%. The subsequent impact will increase the plastic strain energy and crack damage dissipation energy, but the increase in amplitude is small, which is consistent with the change law of the maximum particle displacement, indicating that the multi-particle impact process is a damage accumulation process. It can be seen from the figure that the kinetic energy and elastic potential energy fluctuate after the second impact, and the fluctuations are more pronounced after the third impact. The peaks and valleys of the two correspond to each other. This is because the coating is severely damaged and debris appears, causing collisions and compression between debris and between debris, as well as between debris and the coating, which affect the entire kinetic energy and elastic potential energy in the model.

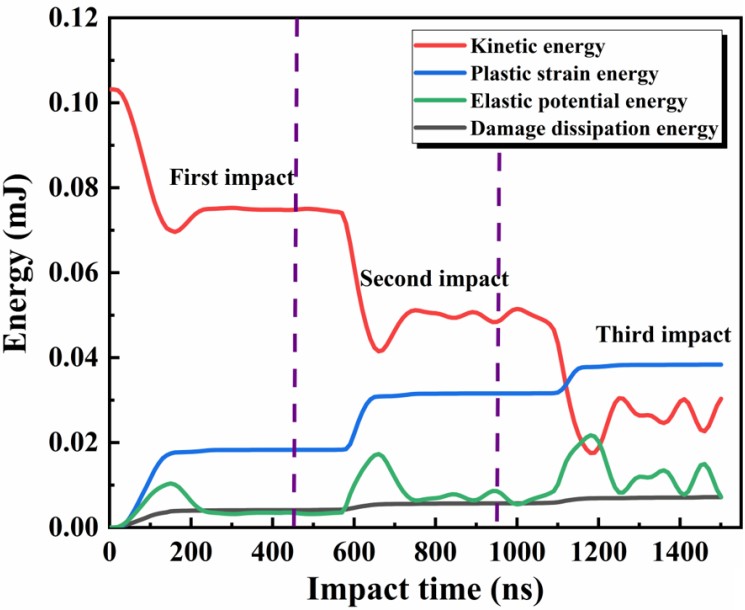

**Figure 5.** Energy variation in TiN/Ti multilayer coating during impact.

Figure 6 shows the coating matrix S11 stress cloud diagram at different moments during the multi-particle impact process (the initial contact, the beginning rebound, and the particle detachment of each particle impact). S11 stress can reflect the in-layer stress state of the material under impact. It can be seen that, in each impact, the maximum tensile stress in the coating also increases to its maximum value with the downward movement of the particle to the lowest position, while the maximum tensile stress gradually decreases with the rebound of the particles, and the maximum stress of the coating increases with the increase in the number of impacts during the impact process. This is because during the impact process, a part of the kinetic energy of the particles is gradually transformed into elastic potential energy, plastic strain energy, and fracture energy. When the particles move to the lowest position, the kinetic energy is the smallest, so the stress in the coating reaches its maximum. When the particles rebound, the elastic potential energy in the coating releases, and the stress in the coating reduces.

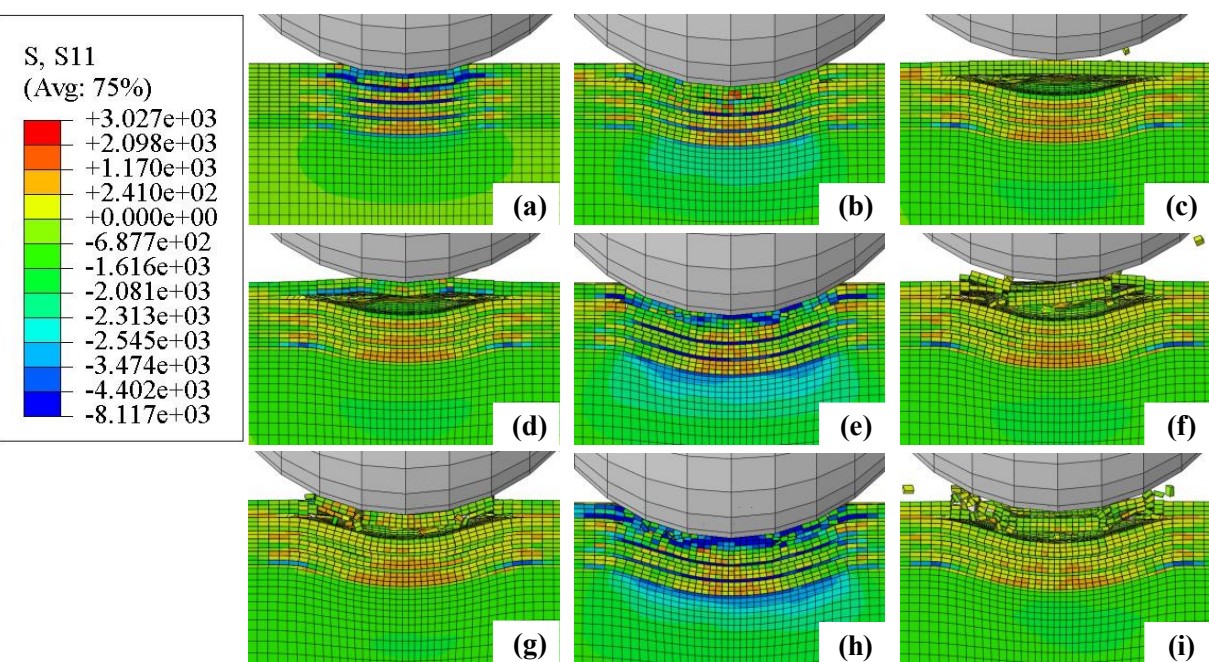

**Figure 6.** S11 stress cloud diagram of TiN/Ti multilayer coating during impact (MPa): (**a**) 50 ns; (**b**) 150 ns; (**c**) 450 ns; (**d**) 500 ns; (**e**) 650 ns; (**f**) 900 ns; (**g**) 1050 ns; (**h**) 1150 ns; and (**i**) 1400 ns.

It can be seen from Figure 6a that, during the first impact, compressive stress appeared on the upper surface of the TiN layer directly below the impact center, while tensile stress appeared on the lower surface of the TiN layer; at the contact edge, tensile stress appeared on the upper surface of the TiN layer, while compressive stress appeared on the lower surface of the TiN layer. When the tensile stress and stress gradient are too high, cracks will occur in the coating, as shown in Figure 3b,c. The coating will produce interface cracks and delamination phenomenon. Under the second and third impacts, the increase in tensile stress and the decrease in erosion resistance due to damage to the coating will also aggravate the impact damage. The initiation and propagation of cracks in the coating will also intensify. Spalling and debris occur when the cracks are connected to each other, as shown in Figure 6f,i.

### 3.1. Damage Evolution during Multi-Particle Impact Process

The output results of SDEG and JCCRT in the post-processing of the ABAQUS software indicate the damage of brittle materials and plastic materials, respectively, and the process of SDEG and JCCRT, increasing from 0 to 1, which indicates the process of material damage to complete failure. Figure 7 shows the characteristic cloud map of coating damage when multilayer coating is impacted. It can be seen that obvious edge cracks were generated at the particle contact edge, where the stress gradient of the coating was large and the tensile stress on the coating surface was high. At the same time, the coating produced interface cracks and longitudinal cracks under particle impact, and longitudinal cracks deflected and stopped at the interface, which indicates that the interface has the ability to hinder crack propagation. Figure 8 shows the stiffness degradation cloud image of the coating at different times during the particle impact process. It can be seen from the stiffness degradation cloud image that the coating cracking after impact is a gradual accumulation process. Firstly, micro-cracks occur in the TiN layer at the top of the coating in the contact area. As the impact progresses, the damage and failure of the Ti layer and the generation of interface cracks lead to delamination, while there are fewer cracks in other layers. After the first impact, serious failure and delamination of the coating occur only in the top Ti layer, while the subsequent impact increases the number of cracks in the impact area and

the number of cracks in other layers. When the cracks expand and connect with each other, serious cracking and massive stripping occur.

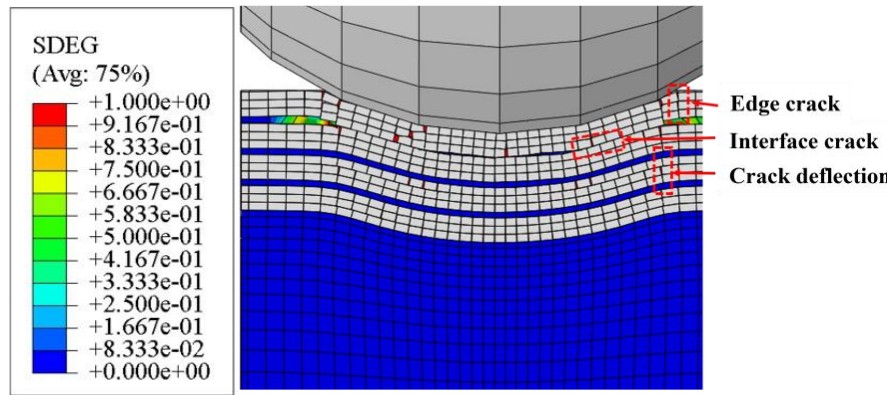

**Figure 7.** Damage cracking morphology of TiN/Ti multilayer coating during impact.

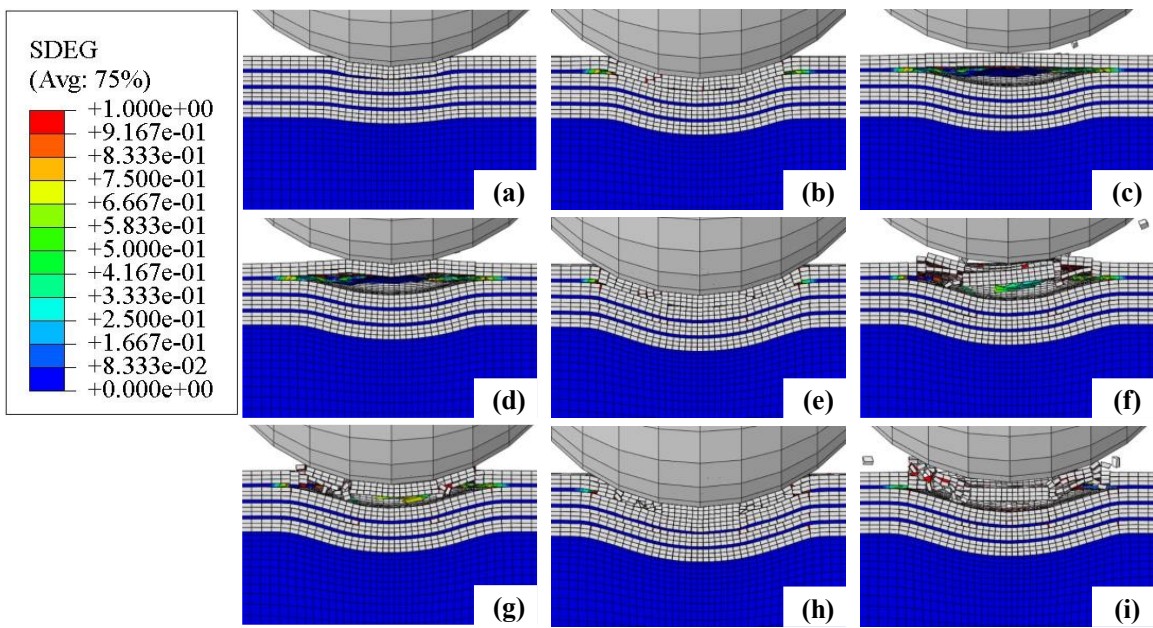

**Figure 8.** Stiffness degradation (SDEG) cloud of TiN/Ti multilayer coatings during impact: (**a**) 50 ns; (**b**) 150 ns; (**c**) 450 ns; (**d**) 500 ns; (**e**) 650 ns; (**f**) 900 ns; (**g**) 1050 ns; (**h**) 1150 ns; and (**i**) 1400 ns.

During the impact process, the coating bears the main deformation, and plastic strain mainly occurs in the top layer of Ti, decreasing along the impact direction. The plastic strain occurring in the matrix is the smallest, indicating that multilayer coatings can effectively protect the matrix and reduce deformation. From the figure, it can be seen that the material in the impact area of the top Ti layer has completely failed due to excessive deformation. In the same layer of Ti, plastic strain was mainly concentrated at the midpoint of the impact center and contact edge. Correspondingly, plastic damage was more severe at this point. When the plastic strain value is too large, the material fails. As shown in Figure 9, ductile metal damage in the Ti layer preferentially occurs at the location of plastic strain concentration.

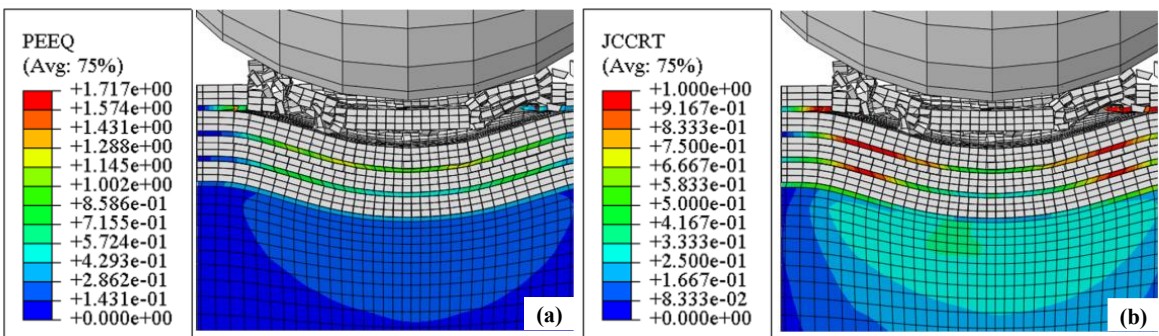

**Figure 9.** Plastic strain in coating (**a**) and ductile metal damage cloud (**b**).

Figure 10 shows the stiffness degradation cloud diagram of the top TiN layer during the impact process, where the damage at the impact pit was the most severe, and lesser damage appeared in other areas of the coating. In the first impact, the damage to the coating intensified, and the damage area at the impact pit significantly expanded. However, after multiple impacts, the area of the damage area remained basically unchanged, and it can be seen from the figure that the element deletion phenomenon occurred at the impact crater, and two obvious circular cracks appeared after the third impact. Figure 11 shows the damage cloud diagram of the top interface and the bottom interface. It can be seen that the material failure occurred at the impact pit of the top layer interface, while the damage in other areas was relatively small. The bottom interface showed obvious deformation in the impact area and formed a circular pit shape. And no material failure occurred, although the damage was serious.

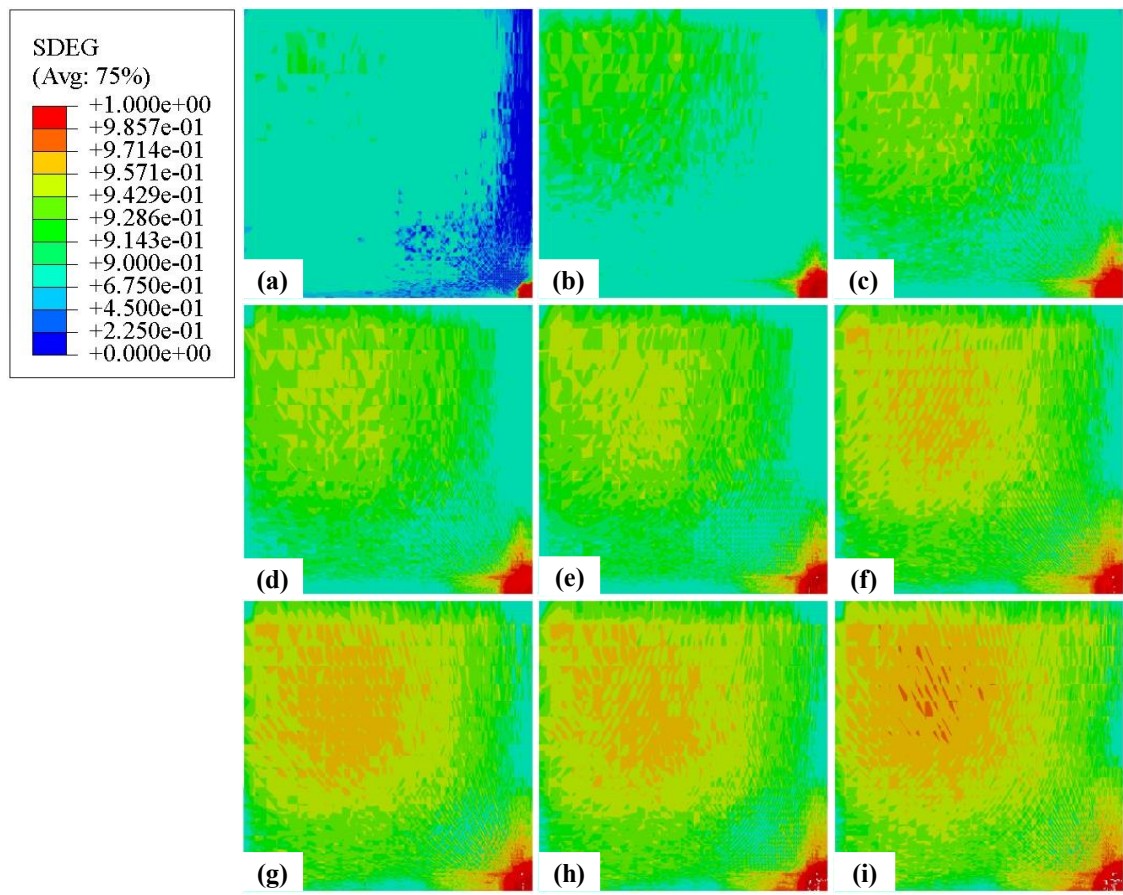

**Figure 10.** Local stiffness degradation of the top TiN/layer during impact (SDEG) clouds: (**a**) 50 ns; (**b**) 150 ns; (**c**) 450 ns; (**d**) 500 ns; (**e**) 650 ns; (**f**) 900 ns; (**g**) 1050 ns; (**h**) 1150 ns; and (**i**) 1400 ns.

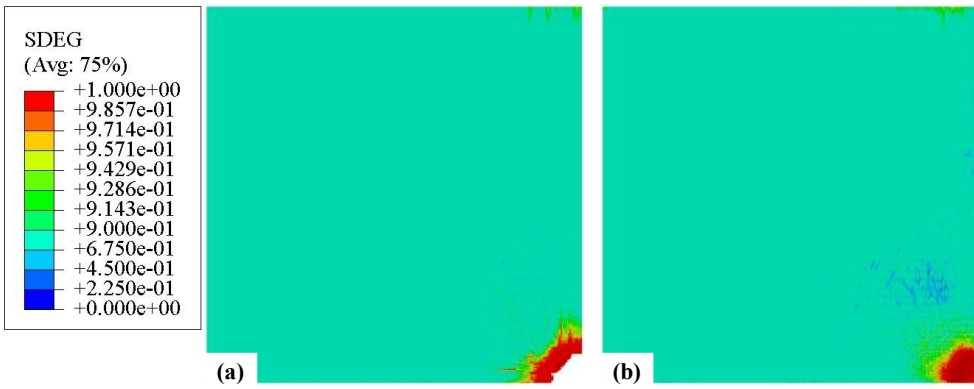

**Figure 11.** Local stiffness degradation (SDEG) cloud of the top interface (**a**) and bottom interface (**b**) after impact.

### 3.2. The Influence of Impact Angle on Erosion Resistance Performance

Figure 12 shows the S11 stress cloud map of the coating substrate under different impact angles (15°, 30°, 45°, 60°, 75°, and 90°) at an impact speed of 100 m/s. The contact area between particles and coatings expands with the increase in impact angle, and the deformation between the coating and substrate increases. At an impact angle of 90°, the extrusion and accumulation of the material occurred at the edge of the impact pit. When the impact angle is higher than 60°, obvious cracks appear in the coating, and a small amount of Ti layer failure occurs. When the impact angle is 90°, the coating damage is most severe. As the impact angle increases, the asymmetry of the stress field distribution gradually weakens, and the stress cloud map at 90° shows a symmetrical shape.

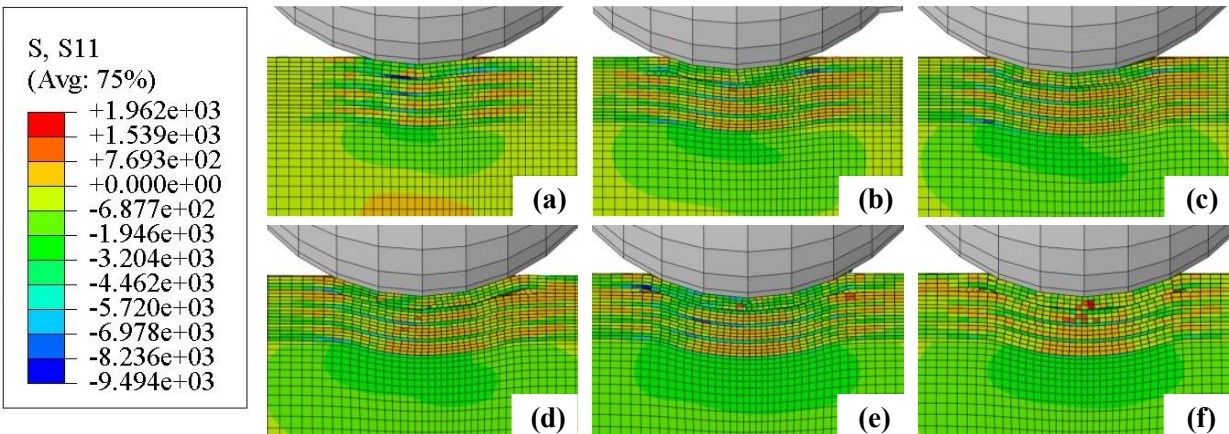

**Figure 12.** S11 stress distribution of TiN/Ti multilayer coating under different angle impacts (MPa): (**a**) 15°; (**b**) 30°; (**c**) 45°; (**d**) 60°; (**e**) 75°; and (**f**) 90°.

It can be seen from the legend that the overall rule of maximum S11 stress increases with the increase in angle because according to the momentum relationship in the impact process [33], the velocity is divided into tangential velocity and normal velocity, and the contact force can be decomposed into tangential contact force and normal contact force. If the velocity is kept constant, during the process of increasing the impact angle to 90°, the tangential contact force and tangential velocity decrease, while the normal contact force and normal velocity increase. Therefore, the contact force acting on the coating surface reaches its maximum under 90° impact. After studying the effect of impact angle on erosion, Chen W et al. [34] found that increasing the impact angle exacerbates the impact damage of the coating, reaching its maximum at a 90° impact angle.

Figure 13 shows the energy variation in the entire model after impact at different angles. The initial total kinetic energy is the sum of the kinetic energy of three particles, which is 0.103 mJ. It can be seen from the figure that as the impact angle increases, the kinetic energy gradually decreases, while the elastic potential energy, plastic strain energy, and crack damage dissipation energy increase. This indicates that the larger the impact angle, the more serious the damage of particles to the coating, and the more serious the damage degree of the coating. When the impact angle is less than 60° under small-angle impact, the kinetic energy and plastic strain energy change almost linearly, while under large-angle impact, the amplitude of the change in kinetic energy and plastic strain energy decreases. At the same time, when the impact angle is greater than 75°, the plastic strain energy exceeds the kinetic energy, and the elastic potential energy exceeds the crack damage dissipation energy. The proportion of plastic strain energy increases with the increase in impact angle, and the increase in crack damage dissipation energy is relatively small, indicating that the plastic strain of the coating plays a major role in energy conversion during the impact process. Improving the plastic toughness and strength of the coating is beneficial for improving its erosion performance.

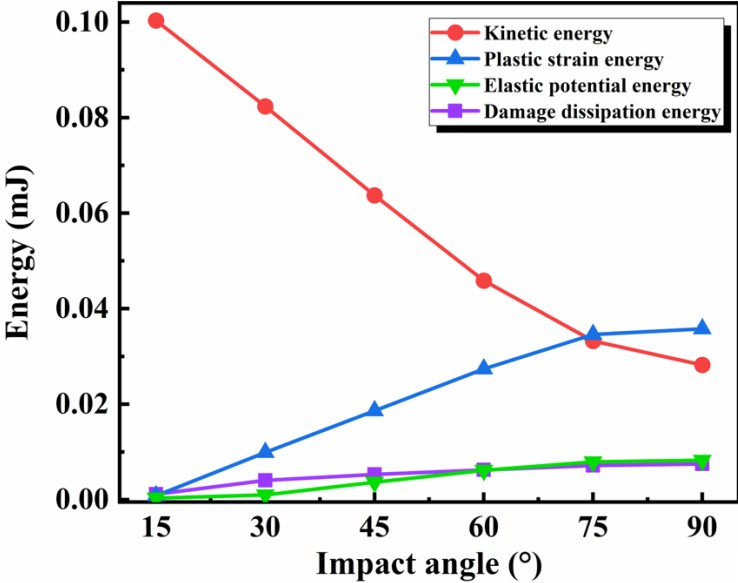

**Figure 13.** The energy of the model after impact at different angles.

## 4. Conclusion

This study focused on TiN/Ti multilayer coatings, the ABAQUS software was used to simulate their multi-particle impact process and to investigate stress–strain distribution and damage law, as well as the influence of impact angle on the erosion resistance of the coating. The results can be concluded as follows:

(1) During the impact process, the stress in the coating will first increase and then decrease, and subsequent impacts will increase the stress in the coating. During the impact process, compressive stress appears on the upper surface of the TiN layer directly below the impact center, while tensile stress appears on the lower surface of the TiN layer. At the impact contact edge, tensile stress appears on the upper surface of the TiN layer, while compressive stress appears on the lower surface of the TiN layer.

(2) During the impact process, the kinetic energy first decreases and then increases, while the elastic potential energy first increases and then decreases. The plastic strain energy and crack damage dissipation energy first increase and then remain stable. And the plastic strain energy accounts for 58.9% of the kinetic energy consumed after the first impact.

(3) In the same Ti layer, plastic strain and ductile metal damage are mainly concentrated at the midpoint between the impact center and the contact edge. The damage to the top layer TiN and interface is relatively severe.

(4) The maximum S11 stress and plastic strain energy in the coating increases with the increase in impact angle. The increase in impact angle aggravates the damage to the coating, and the degree of coating cracks and material delamination intensify.

**Author Contributions:** Z.Y. (Zhanwei Yuan): resources, project administration, writing—review and editing, supervision, and data curation. W.S.: software, investigation, methodology, and writing—original draft. G.H.: writing—review and editing. Y.C.: writing—review and editing. Z.Y. (Zhufang Yang): writing—review and editing. M.G.: writing—review and editing. All authors have read and agreed to the published version of the manuscript.

**Funding:** The authors are very grateful for the support received from Fundamental Research Funds for the Central Universities, CHD (No. 300102311403).

**Institutional Review Board Statement:** Not applicable.

**Informed Consent Statement:** Not applicable.

**Data Availability Statement:** The authors do not have permission to share data.

**Conflicts of Interest:** The authors declare no conflict of interest.

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
