# Peer review of "Simulation of TiN/Ti Multilayer Coating under the Impact of Multiple Particles Based on Cohesive Element"

_coatings, doi:10.3390/coatings14040470_

Round 1

Reviewer 1 Report

Comments and Suggestions for Authors

This work deal with the simulation of damage behavior and erosion resistance of TiN/Ti multilayer coating under multiparticle impact.

The topic is nicely processed. The authors demonstrated their knowledge of simulating processes, but I have a few questions about the work in question:

-       What is the use of that result, since erosion resistance is a very complex problem?

-       How thick was the coating?

-       What was the grain size of the sand and dust particles?

-       In line 105 it is written that "TC4 titanium alloy material was selected as the matrix material", but in table 1 Ti-6Al-V is listed. Which material was chosen as matrix material?

-       Can the authors verify the results of the simulation in practice?

Reviewer 2 Report

Comments and Suggestions for Authors

Article: Simulation of TiN/Ti multilayer coating under the impact of 2 multiple particles based on cohesive element

Comments for Authors: The article I reviewed: "Simulation of TiN/Ti multilayer coating under the impact of 2 multiple particles based on cohesive element" it provides an important topic of multilayer coating, especially with regard to the use of new coating material solutions, and its simulation research at erosion resistane aspect. The article is well written and carefully edited, but has some shortcomings and needs minor improvements.

1. In general, the article is edited well and with great care. The authors clearly indicated the purpose of the research.

2. Lines 91-94: Please provide the actual grid dimensions adopted, both in and out of the particle contact zone. Please explain on what basis these dimensions were chosen. How does this affect the simulation results obtained?

3. Lines 99-101: Signs are appearing: C3D8, COH3D8 and C3D4, and others of this type in the text of the article - please complete the article with nomenclature and explanation of these designations in order to describe the designations clearly and unambiguously.

4. Please replace Ti with Al2O3 in Table 1.

5. Editing - caption of Table 2 separated from the table.

6. The description of the results obtained is very rich. However, some reference to the phenomenon of adhesion is missing. Was such a phenomenon observed at any stage of the research? Please respond to the ongoing research in the context of the phenomenon of adhesion.

7. Editing - caption of Figure 7 separated from the figure.

8. Figure 10 - no description of boxes a to i. There is also a lack of references to these fields in the text, which makes it difficult to read the content.

9. Figure 12 - no description of boxes a to f. There is also a lack of references to these fields in the text, which makes it difficult to read the content.

10. Figure 3 - legends are unreadable

11. Figures: 6, 8, 9, 10, 11 and 12: legends are unreadable. Furthermore, why was the same scale not used in the legends for the individual figures? This would definitely improve the comparative analysis and readability of the figures.

Round 2

Reviewer 1 Report

Comments and Suggestions for Authors

The publication can be published in this version without further additions.